# Morphology Effect of Bismuth Vanadate on Electrochemical Sensing for the Detection of Paracetamol

**DOI:** 10.3390/nano12071173

**Published:** 2022-04-01

**Authors:** Ying Liu, Xiaocui Xu, Churong Ma, Feng Zhao, Kai Chen

**Affiliations:** Guangdong Key Laboratory of Optical Fiber Sensing and Communications, Institute of Photonics Technology, Jinan University, Guangzhou 510632, China; liuying@jnu.edu.cn (Y.L.); szs1997@stu2019.jnu.edu.cn (X.X.); churongma@jnu.edu.cn (C.M.); fzhao@jnu.edu.cn (F.Z.)

**Keywords:** BiVO_4_, morphology control, electrochemical sensor, paracetamol

## Abstract

Morphology-control, as a promising and effective strategy, is widely implemented to change surface atomic active sites and thus enhance the intrinsic electrocatalytic activity and selectivity. As a typical n-type semiconductor, a series of bismuth vanadate samples with tunable morphologies of clavate, fusiform, flowered, bulky, and nanoparticles were prepared to investigate the morphology effect. Among all the synthesized samples, the clavate shaped BiVO_4_ with high index facets of (112), (301), and (200) exhibited reduced extrinsic pseudocapacitance and enhanced redox response, which is beneficial for tackling the sluggish voltammetric response of the traditional nanoparticle on the electrode surface. Benefiting from the large surface-active area and favorable ion diffusion channels, the clavate shaped BiVO_4_ exhibited the best electrochemical sensing performance for paracetamol with a linear response in the range of 0.5–100 µmol and a low detection limit of 0.2 µmol. The enhanced electrochemical detection of paracetamol by bismuth vanadate nanomaterials with controllable shapes indicates their potential for applications as electrochemical sensors.

## 1. Introduction

Drug detection is essential in the monitoring of drug molecules in bio-fluids and plays an important role in drug quality control [1,2]. Paracetamol, also known as acetaminophen, is one of the most popular analgesics/antipyretics and has been applied in effective treatment of pain and fever in adults and children [3,4]. Paracetamol distributes rapidly after oral administration and is easily excreted in the urine. Unlike other analgesic drugs, paracetamol does not produce gastrointestinal damage or untoward cardiorenal effects [5,6]. However, the hypersensitivity or overdose of paracetamol can lead to formation of some liver and nephrotoxic metabolites, such as acute liver necrosis [7]. Moreover, the hydrolytic degradation product of paracetamol is 4-amino-phenol that can be found in pharmaceutical preparations and can cause teratogenic effect and nephrotoxicity [8].

It is desirable to develop an efficient electrochemical catalyst for paracetamol for the quality control of pharmaceuticals, physiological function, and diagnosis in clinical medicine [9]. Semiconductors have been taken as effective photocatalytic and electrochemical sensors for direct detection of paracetamol [10,11,12,13]. Therein, transition metal oxide BiVO_4_, with an excellent charge transport property (hole diffusion length L_p_ = 70 nm) [14,15], has emerged as a highly promising electrocatalytic material with good chemical stability, environmental inertness, and low cost [16,17]. Medeiros et al. reported that BiVO_4_ nanoparticles could be used as a highly efficient and sensitive photoelectrochemical sensor for paracetamol detection [18]. Generally speaking, morphology optimization can further enhance the electrocatalytic performance of material oxides. Control of the size and shape of material oxides is essential to optimize their active areas and favorable ion diffusion channels [19]. As a result, many efforts have been made to engineer metal oxides on the nanoscale that have led to the understanding of their fundamental size- and shape-dependent properties [20]. For example, porous BiVO_4_ with a larger surface area and more reactive sites compared with nanoparticle shape could afford a faster electron transfer rate, as well as higher stability and reproducibility of the sensor [21]. The ability to control the particle morphology can provide a means to tune so-called structure-sensitive catalytic reactions [22,23]. It is highly desirable to be able to synthesize electrocatalytic materials with different morphologies in a facile and controllable manner for the purpose of improving the sensitivity of paracetamol detection, as well as investigating the morphology effect on the electrochemical sensing.

In this work, the BiVO_4_ electrodes with different morphologies were prepared for the purpose of investigating the influence of morphology on sensitivity of electrochemical sensing. The BiVO_4_ samples were synthesized by the microwave approach at 180 °C for 30 min. The addition of additives and the adjustment of the pH value of solutions could change the morphology of the BiVO_4_ samples effectively. Further, the influence of different morphology of BiVO_4_ on the electrochemical detection of paracetamol was explored. Among all synthesized BiVO_4_ samples, the clavate morphological of the BiVO_4_ electrode exhibited the best electrochemical sensing performance on paracetamol with the widest linear detection range (0.5–100 µM) and lowest detection limit (0.2 µM).

## 2. Experimental Section

### 2.1. Apparatus

A microwave chemistry working platform (Model: TOPEX+, PreeKem Scientific Instruments Co., Ltd., Shanghai, China) and high-performance liquid chromatography (Model: Essentia CTO-16L, Shimadzu, Shanghai, China) with a column of 2.1 mm × 10 cm (Model: ZORBAX SB-C18, Agilent, CA, USA) were used. Scanning electron microscopy (SEM) images were obtained by Apreo (Thermo Scientific, Waltham, MA, USA). Field emission transmission electron microscopy (TEM) images were recorded using a TecnaiTM G2 F30 (FEI Co. Ltd., Hillsboro, OA, USA). X-ray diffraction (XRD) characterization was carried out on a D2 PHASER (Bruker, Karlsruhe, Germany) with Cu-Kα as the radiation source (λ = 0.154 nm). X-ray photoelectron spectroscopy (XPS, K-Alpha^+^, Thermo Scientific, MA, USA) was used. The C1s binding energy of adventitious carbon contamination with 284.6 eV was selected as the reference. UV–Vis diffuse reflectance spectra were recorded with a UV–Vis spectrophotometer (Model: Frontier, PerkinElmer Inc., MA, USA). All the electrochemical experiments were carried out using an electrochemical workstation (CH Instrument 660E, Shanghai Chenhua Instrument Co., Ltd., Shanghai, China). The electrochemical experiment was performed using a conventional three-electrode system with the prepared BiVO_4_ as the working electrode, a graphite rod as the counter electrode, and a saturated Ag/AgCl electrode as the reference electrode.

### 2.2. Chemicals and Reagents

Bi(NO_3_)_3_.5H_2_O (99.0%), NH_4_VO_3_ (99.9%), KCl (99.8%), NaH_2_PO_4_ (99.0%), Na_2_HPO_4_ (99.0%), and Na_3_PO_4_ (96%) were obtained from Aladdin Reagent (Shanghai) Co., Ltd., Shanghai, China. HCl, Na_2_CO_3_ (99.5%), and NaOH (97%) were purchased from Macklin (Shanghai) Biochemical Technology Co., Ltd., Shanghai, China. The standard drug of paracetamol tablets (over the counter, OTC) was obtained from Taiji Pharmaceutical Industrial Co., Ltd., Sichuan, China. Phosphate buffered saline (PBS) was prepared (lab temperature at 26 ± 2 °C) by 0.010 M NaH_2_PO_4_, 0.010 M Na_2_HPO_4_, and 0.050 M KCl. All solutions were prepared using ultra-pure water supplied by a Milli-Q system (Millipore, Burlington, MA, USA) with a resistivity of 18.2 MΩ cm.

### 2.3. Preparation of Bismuth Vanadate

The amounts of 5 mM Bi(NO_3_)_3_ and 5 mM NH_4_VO_3_ were dissolved in 10 mL ultra-pure water. Different amounts of NaH_2_PO_4_, Na_2_HPO_4_, Na_3_PO_4_, and Na_2_CO_3_ were used as additives. The amount of 1.0 M NaOH and 36% HCl solutions were used to adjust the pH values of solutions. Then 20 mL amounts of prepared solutions with different pH values were transferred to 100 mL Teflon reactors. The microwave reaction was carried out at 180 °C for 30 min. Different morphologies of BiVO_4_ powders were collected by centrifugation at 12,000 rpm and dried in an oven at 80 °C for 24 h. The BiVO_4_ electrodes with different morphologies were prepared by the spin coating method; 5 mg BiVO_4_ powder was dispersed in 1 mL DI water by ultrasound for 10 min. Then the BiVO_4_ suspension was transferred to FTO substrates by spin coating. The as-prepared BiVO_4_/FTO electrodes were annealed in a muffle furnace with 200 °C for 2 h for stabilization. Finally, the BiVO_4_/FTO electrodes with different morphologies were obtained.

### 2.4. Determination of Paracetamol in Tablets

High performance liquid chromatography (HPLC) was used for the estimation of the content of paracetamol in standard drug of paracetamol tablets. The method was carried out on a Hichrom C18 (25 cm × 4.6 mm i.d., 5 µm) column with a mobile phase consisting of methanol and icy ultra-pure water containing formic acid (volume ratio of 50%/49.9%/0.1%) at a flow rate of 0.2 mL min^−1^. Detection was carried out at 257 nm. Standard stock solutions of 0, 10, 20, 50, and 100 µmol of paracetamol were prepared in PBS (pH = 7.4), respectively. The injection volume of solution was 50 μL.

## 3. Results and Discussion

### 3.1. Characterization of BiVO_4_ with Different Morphologies

The synthesis procedures with different conditions to obtain different morphologies of BiVO_4_ are summarized in Appendix A. The morphology of bismuth vanadate could be controlled with different additives, including NaH_2_PO_4_, Na_2_HPO_4_, Na_3_PO_4_, and Na_2_CO_3_. The pH values of the solutions were adjusted in the range of 2 to 10 with HCl or NaOH. Due to the differences in ionization and hydrolysis of phosphate, carbonate, and alkaline, the morphology of bismuth vanadate could form nanosheet, tetrahedron, cuboid, sphere, or irregular shapes. In this article, several representative morphologies of bismuth vanadate were selected, which were denoted as clavate, fusiform, flowered, bulky, and particle BiVO_4_, as illustrated in Figure 1. Certain bismuth vanadate samples were chosen as representatives to discuss the effect of morphology on electrochemical properties of BiVO_4_ materials.

The particle BiVO_4_ was obtained by adding Na_3_PO_4_ to the precursor solution (Figure 1E). Then HCl was added drop by drop to adjust the pH value to 4. The Bi^3+^ ions and VO_3_^−^ ions combined to form nanoparticles. The size of BiVO_4_ nanoparticles ranged from 30 to 300 nm. The clavate and fusiform topography were obtained by adding Na_2_HPO_4_ (Figure 1A,B). The pH values of solutions were adjusted to 9.5 and 3.6, respectively. The pH value of 0.1 M Na_2_HPO_4_ solution was about 9 (Ka_1_ = 7.1 × 10^−3^). Due to the similar pH values of solution, the clavate morphology BiVO_4_ was formed directly and rapidly. By contrast, the fusiform BiVO_4_ was formed by the aggregation of nanoparticles (Appendix A), which was due to the partial dissolution and structural reorganization during the process of adjusting the pH value of the solution. The flowered BiVO_4_ was formed with the addition of NaH_2_PO_4_. NaH_2_PO_4_ solution is acidic (Ka_1_ = 6.2 × 10^−8^). NaOH was added slowly to change the pH value of solution to 7.7. The BiVO_4_ gradually assumed a cross-linked flowered-sphere structure with a diameter of ca. 5 µm (Figure 1C). The bulky topography of BiVO_4_ could be obtained by adding NaHCO_3_ and adjusting the pH value to 6 (Figure 1D). The bulky BiVO_4_ showed a cube structure with a hole in the center.

Due to the different hydrolysis and ionization rates of carbonate and phosphate, the pH values of the solution were different, which further affected the nucleation rate of the Bi^+^ in solution. A further change in the pH of the solution would lead to the dissolution or reshaping of BiVO_4_ samples. As a result, the morphology control of BiVO_4_ could be achieved with different additives and pH values.

The selected BiVO_4_ samples were characterized with a powder X-ray diffractometer (Figure 2A). Bulky BiVO_4_ exhibited reflection planes (101), (200), (112), and (312) corresponding to the 2θ values of 18.3°, 24.3°, 32.7°, and 48.4°, respectively. These values are well matched with standard JCPDS file No. 14-0133, illustrating a tetragonal phase of bulky BiVO_4_. The clavate BiVO_4_ sample exhibited a tetragonal structure. Moreover, diffraction peaks of impurities (BiO_2_) were observed in clavate BiVO_4_. Particle BiVO_4_ exhibited sharp diffraction peaks at 18.6°, 28.6°, and 30.9°, which correspond to (101), (112), and (004) crystal planes, respectively (JCPDS file No. 48-0744). The fusiform morphological sample exhibited (011), (004), and (113) planes of orthorhombic structure as indicated by the 2θ of 19.1°, 29.8°, and 33.1° (JCPDS file No. 12-0293). Notably, there were no clear diffraction peaks in the flowered BiVO_4_ sample, indicating poor crystallization of the flowered BiVO_4_ sample. Overall, the selected BiVO_4_ samples exhibited different properties, which are summarized in Table 1.

The surface chemical states of selected BiVO_4_ samples were further investigated by XPS. As for the O 1s spectrum (Figure 2F), the peaks around 529.2 eV were clearly shown in particle, bulky, and flowered BiVO_4_ samples, which could be attributed to the lattice oxygen (O^2−^) in BiVO_4_ [24]. High-resolution XPS spectra of O 1s after peak fitting were shown in Appendix A. The peak at 529.2 eV was not observed in clavate and fusiform BiVO_4_ samples, while a new peak at 530.4 eV appeared, which could be ascribed to the lattice oxygen in bismuth oxide, suggesting the surface of clavate and fusiform BiVO_4_ were oxidized. The existence of the surface oxide layer in clavate and fusiform BiVO_4_ samples was further confirmed by high-resolution XPS spectra of Bi 4f (Figure 2E) and V 2p (Figure 2D). The Bi 4f of particle, bulky, and flowered BiVO_4_ samples showed two characteristic peaks at 164.8 eV and 159.1 eV that were attributed to Bi 4f_5/2_ and Bi 4f_7/2_, respectively. The Bi 4f of clavate and fusiform BiVO_4_ samples showed an apparent shift to high binding energy, attributed to the bismuth oxide. A similar phenomenon could be observed in the V 2p spectra. The V 2p peaks were not shown in clavate and fusiform BiVO_4_ samples due to the surface oxide layer that covered the single V orbit. XRD results show that the clavate and fusiform BiVO_4_, which were both synthesized with Na_2_HPO_4_, were much more unstable. The VO_4_ unit in BiVO_4_ easily formed a new bismuth oxide unit as well as oxygen vacancies on the surface of BiVO_4_.

### 3.2. Optical Analysis of BiVO_4_ Samples

The BiVO_4_ powders obtained from different precursors showed different colors (the inset of Figure 2B). The flowered, bulky, and particle BiVO_4_ showed yellow color with the band edge of optical absorption at around 550 nm. The clavate and fusiform BiVO_4_ exhibited a white color with a shift of optical absorption edge to about 400 nm (Figure 2B). The change of the color was due to the surface oxide layer of BiVO_4_. The UV–vis DRS data were combined with the Kubelka–Munk (K–M) relation to study the association of diffused reflectance with the absorption coefficient: F(R) = (1 − R)^2^/2R, where F(R) is the Kubelka–Munk function, and R is the absolute reflectance of the sample. The optical band gap of the prepared samples is calculated using Tauc’s equation: F(R) hυ = A(hυ − E_g_)^n^, where n = 2 for a directly allowed transition, and n = 1/2 for an indirectly allowed transition, and A is a constant and hυ is photon energy [25]. According to the calculation, the band gap (E_g_) of the selected BiVO_4_ was summarized in Table 1. The band gap of intrinsic BiVO_4_ is around 2.4 eV [26]. The particle, bulky, and flowered BiVO_4_ showed a comparable value of E_g_ with intrinsic BiVO_4_, while clavate and fusiform BiVO_4_ had a larger value of E_g_ due to the surface oxide layer. Generally speaking, BiVO_4_ shows excellent photoelectrochemical performance due to its suitable band gap. To investigate the influence of the solar light in photo-assisted detection of paracetamol, differential pulse voltammetry (DPV) of clavate BiVO_4_ on determination of paracetamol in the dark and under illumination was studied. As displayed in Appendix A, the photoresponse current on BiVO_4_ electrode increased from 17.5 to 21.0 mA cm^−2^, which indicates that solar light has a positive effect on improving the sensitivity of paracetamol detection. 

### 3.3. Electrochemical Response at Various BiVO_4_ Electrodes

Electrochemical techniques have been widely explored in the detection of paracetamol in biological fluids and tablets due to their simple pretreatment procedure, high sensitivity, low time of analysis, and low costs over other analytical methods [27,28]. To understand the effect of morphology control on electrochemical performance, different morphological BiVO_4_ were examined in 0.01 M PBS with and without 100 µM paracetamol using CV. All the electrochemical experiments were carried out under dark conditions in order to exclude the influence of other factors.

Figure 3A–E shows the CV response of the paracetamol re-dox process on different BiVO_4_ electrodes. The CV curves of all the BiVO_4_ electrodes displayed a strong anodic peak (E_pa_) at 0.48 V. The different peaks at BiVO_4_ electrodes could be observed in the process of backward scan with two small cathodic peaks (E_pc_), which were registered at 0.4 and 0.1 V, respectively. The chemical reaction was coupled to the electrochemical product for the oxidation of acetaminophen, i.e., *N*-acetyl-*p*-benzoquinoneimine (NAPQI) [29]. The competition between two forms of NAPQI (protonated and unprotonated species) has been proposed by Kissinger et al. [30]. The anodic peak currents in CV curves were higher than the cathodic ones, where the most probable coupled chemical reaction was the hydration of the NAPQI molecule, leading to a lower cathodic current. The limited cathodic current illustrates that the reduction of the radical intermediate is controlled by dynamics. In addition, the low cathodic current suggests that the oxidation of paracetamol on the surface of the BiVO_4_ electrode is more efficient than its reduction. 

In order to investigate the influence of morphology on the electrochemical performance, the compared CV results of BiVO_4_ with clavate, fusiform, flowered, and bulky, as well as the reference morphology of particle shape are shown in Figure 3A–E. The reference morphology of particle BiVO_4_ exhibited high electrochemical performance of 75 µA cm^2^ with 100 µM paracetamol when the applied potential was 0.5 V. However, the particle BiVO_4_ showed large capacitance and a sluggish voltammetric response. The pseudocapacitance could affect the redox peak during the voltammetric response, and the extrinsic pseudocapacitance arose at the electrode surfaces along with the gradual BiVO_4_ nanonization [31]. The bulky and flower BiVO_4_ exhibited low current density due to their low specific surface area and poor crystallinity, respectively (Figure 3C,D). It is noted that the fusiform BiVO_4_ with orthorhombic structure and clavate BiVO_4_ with tetragonal structure were both covered by bismuth oxides. However, the fusiform BiVO_4_ (Figure 3B) showed weaker catalytical performance compared with that of clavate BiVO_4_, illustrating the tetragonal structure is beneficial for catalytic performance. Compare to BiVO_4_ with other morphologies, the clavate BiVO_4_ (Figure 3A) exhibited the highest electrochemical performance with a maximum redox peak current of 100 mA cm^−2^ in the 0.1 M PBS solution containing 100 µM paracetamol, owing to the large specific surface area and surface oxygen vacancies, which could effectively improve the charge transfer characteristics and increase the charge diffusion coefficient of BiVO_4_. Moreover, the clavate BiVO_4_ could afford favorable channels for ion diffusion, which further increased the contact between the carriers and the drug molecules.

In order to suppress the influence of charging current and obtain higher sensitivity, the DPV was introduced into electrochemical analysis. The DPV experimental parameters were optimized at 50 ms pulse width, 50 mV pulse amplitude. The DPV curves of different BiVO_4_ samples in different concentrations of paracetamol solutions are shown in Appendix A, and the electrochemical performances on morphology-dependent BiVO_4_ electrodes in 0.01 M PBS without and with 100 µM paracetamol from DPV results are summarized in Figure 3F. Accordingly, clavate BiVO_4_ with high crystallinity showing the highest electrochemical signal was chosen to systematically investigate its paracetamol sensing performance.

### 3.4. Structural Characterization of Clavate BiVO_4_

TEM characterizations were performed on the clavate BiVO_4_ with the length of several micrometers (Figure 4A,B). Figure 4D,E show the high magnification TEM images of clavate BiVO_4_ that show well-resolved lattice fringes with an interlayer spacing of 0.274, 0.229, and 0.362 nm corresponding to the (112), (301), and (200) planes of tetragonal BiVO_4_ (yellow color), respectively. Previous studies illustrated that BiVO_4_ with high-index planes promotes the catalytic activity compared with low-index (010), (110), and (101) facets [32,33]. Thus, the clavate BiVO_4_ with high index lanes (112), (301), and (200) at the surface could show good electrocatalytic performance. The high index lanes refer to a facet where one of the indexes is greater than 1 in (h, k, l). The lattice fringes with an interlayer spacing of 0.315 nm (white color) corresponded to the (111) crystal plane of BiO_2_, suggesting oxidization at the surface of BiVO_4_. The EDS data in Figure 4C depict the uniform distribution of Bi, V, and O elements. The TEM characterization proved the tetragonal structure of clavate BiVO_4_, and the surface of BiVO_4_ was partially covered by BiO_2_ nanoparticles.

### 3.5. Electrochemical Investigation of Paracetamol on Clavate BiVO_4_

The effect of the clavate BiVO_4_ electrode on the electrochemical detection of paracetamol was further investigated. Cyclic voltammograms at the clavate BiVO_4_ electrode in 100 µM paracetamol with scan rates of 1 to 480 mV s^−1^ are shown in Figure 5A. The anodic peak potentials shifted with the increase of scan rate, and the anodic peak currents presented a linear dependence on the scan rate (Figure 5B). A linear regression equation was adopted for the anodic peak, logI_pa_ = 0.421 logυ + 0.905, giving the correlation coefficient values of R^2^ = 0.999. It demonstrated that oxidation of paracetamol is an irreversible redox process with diffusion-controlled mass transport. The regression equation was obtained as E_pa_ = 0.0493 logυ + 0.404 (R^2^ = 0.996, Figure 5C). The number of electrons involved in the reaction and the charge transfer coefficient could be calculated as 2 and 0.5, respectively. These results indicate that two electrons are involved in the electrochemical redox process of paracetamol. The paracetamol exhibited sluggish voltammetric response at the traditional electrode surface, which restricted the sensitivity of the electrochemical sensor. To solve this problem, DPV was used to improve the detection sensitivity. DPV curves of the clavate BiVO_4_ electrode for paracetamol determination are shown in Figure 5D. The oxidation DPV peak of paracetamol was observed at about +0.35 V on the clavate BiVO_4_ electrode. A linear correlation between the current density and the paracetamol concentration was obtained in the range from 5 × 10^−7^ to 1 × 10^−5^ M, which could be represented by a regression equation as follows: I = 4.65 × 10^−2^ c + 5.51 (R^2^ = 0.997). The detection limit of the sensor was calculated as 2 × 10^−7^ M (S/N = 3). The obtained DPV data revealed that the clavate BiVO_4_ electrode had more competitive analytical performance and a much lower detection limit compared with BiVO_4_ electrodes of other morphologies (Appendix A). The comparisons between the clavate BiVO_4_ electrode and some reported electrodes for paracetamol determination are summarized in Appendix A. The clavate BiVO_4_ electrode exhibited analytical performances with acceptable sensitivity and a wide linear range. To evaluate the selectivity and stability of the clavate BiVO_4_ electrode, metal interference ions such as K^+^, Na^+^, Ni^+^, Zn^2+^, Co^2+^, Mg^2+^, Cd^2+^, Fe^2+^, Pd^2+^, Al^3+^, and Fe^3+^ were each added into a standard solution containing 100 µM paracetamol. As shown in Figure 6A, the addition of 0.10 M metal ion species did not affect the DPV current response of paracetamol on the clavate BiVO_4_ electrode. The result illustrates that the clavate BiVO_4_ electrode has excellent selectivity, even in the presence of a 1000-fold concentration of interference species.

The mechanism of the electrochemical oxidation and reduction of paracetamol is shown in Figure 6B. The applied voltage promotes the separation of electrons and holes in BiVO_4_. The oxidation peak of paracetamol could vary with the pH of the solution. Paracetamol converts to intermediate NAPQI easily when the pH of solution is 7.4. NAPQI can stably exist in the solution in a deprotonated form. The CV of paracetamol shows an oxidation peak and a relatively weak reduction peak. With the progress of the reaction, NAPQI gradually transforms into benzoquinone through other intermediates. The reduction peak of benzoquinone could be observed in CV.

### 3.6. Determination of Paracetamol in Pharmaceutical Samples

The analytical applicability of the clavate BiVO_4_ electrode was tested by determining the concentration of paracetamol in compound paracetamol tablets (Ⅱ) (250 mg per pill). One tablet was dissolved in 0.01 M PBS solution (pH = 7.4). The concentration of paracetamol in the measured solution with a calculated concentration of 45.0 µM was detected by HPLC and DPV methods, respectively. Compared with the HPLC results, the values obtained by the DPV method exhibited high credibility. Moreover, the DPV method showed good reliability and repeatability during three times tests. The recoveries with 96.1–101.9% are given in Table 2, indicating that the fabricated clavate BiVO_4_ sensor is accurate and sensitive enough for detecting paracetamol in pharmaceutical tablets. The above results show that the DPV method is a simple and reliable method to detect the content of paracetamol in drugs. The comparison between our work and some reported sensors for paracetamol determination were shown in Appendix A. The clavate BiVO_4_ electrode shows good stability and reliability in electrochemical detection of paracetamol.

## 4. Conclusions

In summary, the ability to control the morphology of BiVO_4_ has great development prospects in high sensitivity electrochemical analysis. The BiVO_4_ samples with different morphologies were obtained by adding different additives, and changing the pH values of the solution. The electrochemical properties of different morphology BiVO_4_ were systematically studied in this article. The clavate BiVO_4_ with high index lanes at the surface could solve the problem of the sluggish voltammetric response of the traditional nanoparticle on the electrode surface. The clavate BiVO_4_ also could afford favorable channels for ion diffusion, increasing the contact between the carriers and the drug molecules. Therefore, the clavate BiVO_4_ with a tetragonal structure exhibited the highest sensitivity and lowest detection limit among all selected BiVO_4_ electrodes. In the DPV mode, the clavate BiVO_4_ electrode showed linear responses over the concentration range of 0.5–100 µM (R^2^ = 0.998) for paracetamol detection, and the LOD value was found to be 0.2 µM. The study of the relationship between morphology and electrocatalytic performance could provide very important information on the reaction activity and selectivity on target.

## Figures and Tables

**Figure 1 nanomaterials-12-01173-f001:**
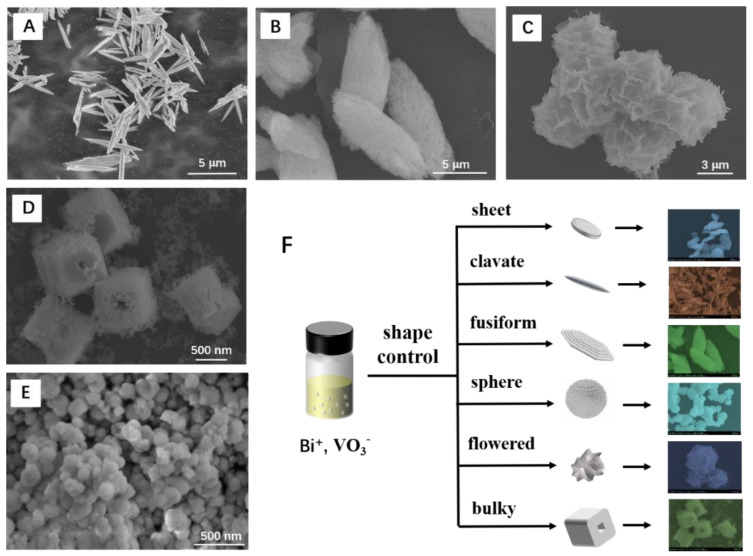
SEM images of BiVO_4_ with selected morphology: (**A**) clavate, (**B**) fusiform, (**C**) flowered, (**D**) bulky, and (**E**) particle. (**F**) Schematic diagram of shape control of the bismuth vanadate.

**Figure 2 nanomaterials-12-01173-f002:**
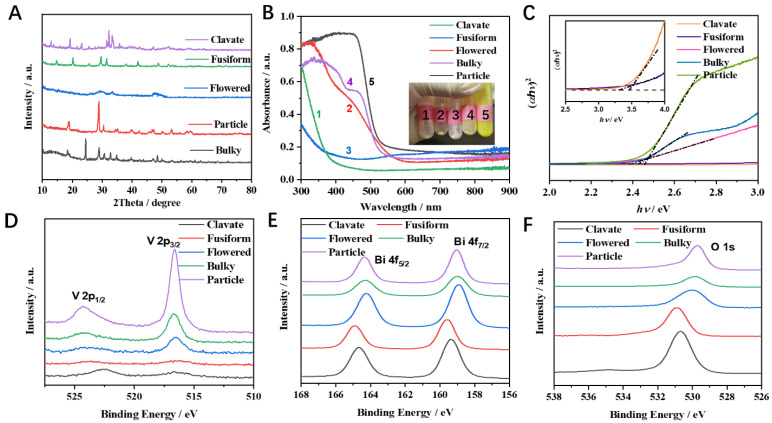
(**A**) XRD patterns, (**B**) UV–Vis spectra, (**C**) Kubelka–Munk plots of BiVO_4_. Inset of (**B**) is the photographs of clavate, fusiform, flowered, bulky, and particle from 1 to 5. High-resolution XPS spectra of (**D**) V 2p, (**E**) Bi 4f, and (**F**) O 1s of different morphologies BiVO_4_.

**Figure 3 nanomaterials-12-01173-f003:**
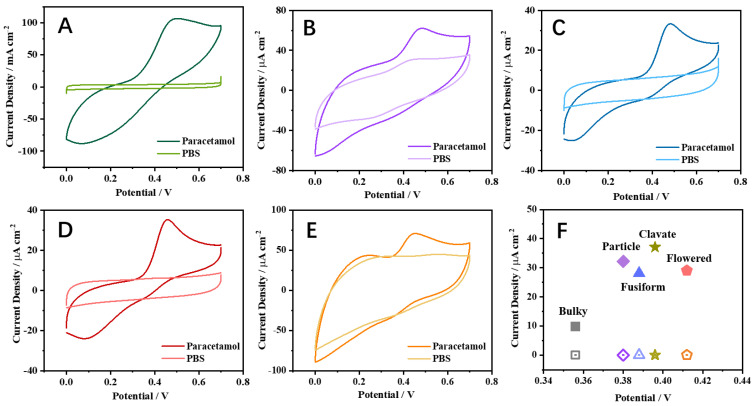
CV curves at (**A**) clavate, (**B**) fusiform, (**C**) flowered, (**D**) bulky, and (**E**) particle BiVO_4_ in 0.01 M PBS with and without 100 µM paracetamol. Scan rate: 100 mV s^−1^. (**F**) The current densities at BiVO_4_ electrode in 0.01 M PBS without (hollow icon) and with (solid icon) 100 µM paracetamol from the DPV curve. The X−axis is the peak potential at different morphologies of BiVO_4_ from DPV curves.

**Figure 4 nanomaterials-12-01173-f004:**
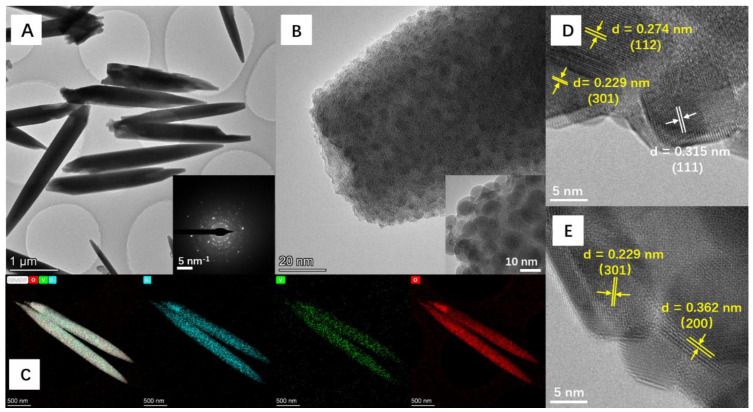
(**A**) TEM, (**B**,**D**,**E**) HRTEM, and (**C**) EDS images of clavate BiVO_4_. Inset of (**A**) is the SAED pattern, and the inset of (**B**) is the magnification of clavate BiVO_4_.

**Figure 5 nanomaterials-12-01173-f005:**
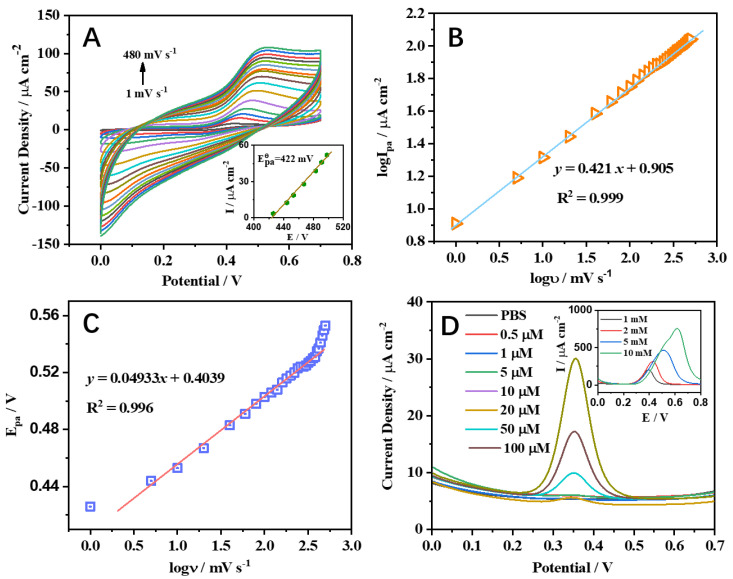
(**A**) CV curves at the clavate BiVO_4_ electrode in 100 µM paracetamol with different scan rates from 1 to 480 mV s^−1^. Inset of (**A**) is a plot of the peak current against peak potential with different scan rates. Dependence of (**B**) logυ−logI_pa_ and (**C**) E_p_−logυ at the clavate BiVO_4_ electrode. (**D**) DPV curves at clavate BiVO_4_ electrode with the concretion of paracetamol from 0 to 100 µM. Inset of (**D**) shows DPV voltammograms corresponding to the high concentration of paracetamol with a 1–10 mM range.

**Figure 6 nanomaterials-12-01173-f006:**
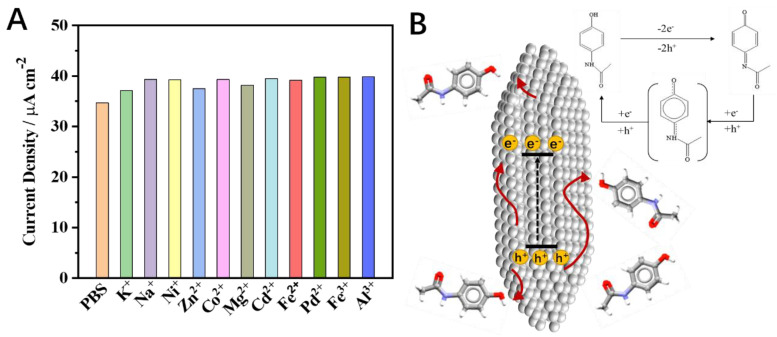
(**A**) Anti-interference experiment of different ions in 100 µM paracetamol with different metal ions. The concentration of the metal ions is 0.10 M. The data of current density are collected by DPV. (**B**) Mechanism of the electrochemical oxidation and reduction of paracetamol.

**Table 1 nanomaterials-12-01173-t001:** Properties of BiVO_4_ samples with different morphologies.

Shape	Structure	Size	Color	Band Gap	Impurity
clavate	tetragonal	L = 5 µm W = 400 nm	white	3.45 eV	bismuth oxide
fusiform	orthorhombic	L = 5–10 µm W = 2–4 µm	off-white	3.30 eV	bismuth oxide
flowered	tetragonal	D = 5 µm	light-yellow	2.38 eV	None
bulky	tetragonal	D = 600 nm	yellow	2.40 eV	None
particle	tetragonal	D = 100–200 nm	yellow	2.45 eV	None

L = length, W = width, D = diameter.

**Table 2 nanomaterials-12-01173-t002:** Analytical application of paracetamol in real samples.

Sample	No.	Calculated/µM	Found by HPLC/µM	Found by DPV/µM	Recovery/%
Paracetamol tablets (Ⅱ)	1	45.0	41.4	42.2	101.9
	2	45.0	40.9	39.6	96.8
	3	45.0	40.8	39.2	96.1

## Data Availability

Data available upon reasonable request to the authors.

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
