# Peer review of "Morphology Effect of Bismuth Vanadate on Electrochemical Sensing for the Detection of Paracetamol"

_nanomaterials, 2022, doi:10.3390/nano12071173_

Round 1
Reviewer 1 Report
The paper can be published after major revision reflecting comments iserted as yelow notes into attached pdf of submitted paper and suplement

Author Response
Thanks very much for giving us the opportunity to revise the article. We have made changes to according to the reviewers' comments, and all the changes are highlight in manuscript.
Question 1: Hollow icons are difficult to see (Page 9).
Answer to Q1: The hollow icons in Fig.3 have been increased bold.
Question 2: Is it a really DPV? The paraments must be given and DPV must be mentioned in the text as well. (Page 9)
Answer to Q2: The current densities of BiVO4 electrode in Fig. 3F were obtained from DPV detection. The data is extracted from the maximum peak in Fig. 5 D and Fig. S4. The DPV parameters have been added in manuscript.
“In order to suppress the influence of charging current and obtain higher sensitivity, the differential pulse voltammetry (DPV) was introduced into electrochemical analysis. The DPV experimental parameters were optimized at 50 ms pulse width, 50 mV pulse amplitude.”
Question 3: Give reference an auxiliary electrode. (Page 4)
Answer to Q3: The reference and counter electrodes were supplied in the manuscript (Page 2).
“The electrochemical experiment was performed using a conventional three-electrode system with the prepared BiVO4 as working electrode, a graphite rod as the counter electrode and a saturated Ag/AgCl electrode as the reference electrode”
Question 4: Fig. 3 at which technique DPV or CV. (Page 10)
Answer to Q4: The Fig. 3A to 3E shows the cyclic voltammetry (CV) test results of different BiVO4 electrode. The Fig. 3F exhibits the current densities of different morphologies BiVO4 electrodes which measured by differential pulse voltammetry (DPY).
Question 5: Dimensions are missing on both axes (Fig.5)
Answer to Q5: The dimensions of x and y axes have been added in Fig.5.
Question 6: Keep in mind that Ip depends on v not vice versa. (Fig. 5)
Answer to Q6: Thanks for reviewer’s suggestion. I change the “log υ vs. log Ipa” to “log υ - log Ipa”.
Question 7: Reasonable number of valid digits should be used both here and in all other relevant cases. (Page 13)
Answer to Q7: Thanks for reviewer’s advice. We already checked all the significant digits in manuscript.
Question 8: English must be improved and more details given. (Page 21)
Answer to Q8: Thanks for reviewer’s suggestion. We changed the figure note of Fig. S1 as “The synthesis produces of morphology-dependent BiVO4 by stirring, hydrothermal and micro wave methods.” And more detail about synthesis process was supplied in SI.
Question 9: It should be explained what dark and light means. (Fig. S5)
Answer to Q9: Thanks for reviewer’s advice. Detailed notes are added in the figure S5. “The DPV test was performance under dark condition (labeled as dark) and in illumination condition (labeled as light).”
Question 10: All abbreviations should be explained in footnote. (Table S1)
Answer to Q10: Thanks for reviewer’s suggestion. The footnote of Table S1 has been added.
LOD: limit of detection; PBS: phosphate buffer saline; BR buffer: Britton-Robison buffer; MWCNTs: multi-wall carbon nanotubes; GO: graphene oxide; PEDOT: Poly(3,4-ethylenedioxythiophene); PS-PNIPAm-PS: polymer poly(styrene-b-(N-isopropylacrylamide)-b-styrene); MWCNTs-COOH: carboxylated multi-walled carbon nanotubes; N-GQDs: amino-functionalized graphene quantum dots.”
Question 11: How probable taken should be given here. (Table 1)
Answer to Q11: The Table 2 was revised according to reviewer’s suggestion. And the calculated concentration of paracetamol in paracetamol tablets was calculated by drug instructions.
We have revised all the tropes in manuscript according to reviewer’s suggestion, including:
Page 2:
- “Morphology” was revised as “morphology”.
- “high index lanes” was revised as “high index planes of (112), (301) and (200).
Page 3:
- “[3, 4]” was revised as “[3, 4]“.
- ”focus“was revised as “attention”.
- “a modest charge transport property” was revised as “an excellent charge transport property”.
- “control” was revised as “control of”.
- “faster electron transfer rate and higher stability, reproducibility of the sensor” was revised as “faster electron transfer rate, as well as higher stability and reproducibility of the sensor”.
Page 4:
- “was” was revised as “were”.
- “Scanning electronic microscopy” was revised as “Scanning electron microscopy”.
- “OCT” was revised as “over the counter, OTC”.
- “Phosphate buffer saline (PBS) of 0.01 M was prepared (lab temperature at 26 ± 2 oC) by NaH2PO4, Na2HPO4 and KCl.” was revised as “Phosphate buffer saline (PBS) was prepared (lab temperature at 26 ± 2 oC) by 0.010 M NaH2PO4, 0.010 M Na2HPO4 and 0.050 M KCl.”
- “ultra-pure water” was revised as “10 mL ultra-pure water”.
- “NaH2PO4, Na2HPO4, Na3PO4 and Na2CO3 were used as additive” was revised as “Different amount of NaH2PO4, Na2HPO4, Na3PO4 and Na2CO3 were used as additive.”.
- “NaOH and HCl” was revised as “1 M NaOH and 36% HCl solution”.
Page 5:
- “2.4 Content measurement of paracetamol tablet” was revised as “Determination of paracetamol in tablets”.
- “High performance liquid chromatography (HPLC) method is sed for the estimation of the content of paracetamol form pharmaceutical dosage forms.” was revised as “High performance liquid chromatography (HPLC) was sed for the estimation of the content of paracetamol in standard drug of paracetamol tablets.”.
- “were” was revised as “are”.
- “which denoted as” was revised as “which are denoted as”.
- “The size distribution of BiVO4 nanoparticles is range from 30 to 300 nm.” was revised as “The size distribution of BiVO4 nanoparticles is from 30 to 300 nm.”.
- “0.1 moL Na2HPO4” was revised as “0.1 M Na2HPO4”.
- “which due to the” was revised as “which is due to the”.
Page 6: “Further change the pH of the solution would leading to the dissolve or reshape of BiVO4 catalyst.” Further change of the pH value of the solution would lead to the disslution or reshape of BiVO4 catalyst.
Page 7: “was” was revised as “were.
Page 8:
1.“XRD results show that the clavate and fusiform BiVO4 which both synthesis with the Na2HPO4 were much more unstable. The VO4 unit in BiVO4 is easily be destroyed and forms a new bismuth oxide unit as well as oxygen vacancies at the surface at BiVO4” was revised as “XRD results show that the clavate and fusiform BiVO4 which were both synthesized with Na2HPO4 were much more unstable. The VO4 unit in BiVO4 easily forms a new bismuth oxide unit as well as oxygen vacancies on the surface of BiVO4.”;
- “yellow-colored” was revised as “yellow color”;
- “direct allowed transition” was revised as “directly allowed transition”.
Page 9:
- “less time-consuming” was revised as “low time of analysis”;
- The title of Fig. 3 was revised as “CV curves at (A) clavate (B) fusiform (C) flowered, (D) bulky and (E) particle BiVO4 in 0.01 M PBS with and without 100 M paracetamol. Scan rate: 100 mV s−1. (F) The current densities at BiVO4 elec-trode in 0.01 M PBS without (hollow icon) and with (solid icon) 100 M paracetamol from DPV curve. The X-axis is the peak potential at different morphologies BiVO4 from DPV curves.”
Page 10:
- “The different peaks of BiVO4 electrodes” was revised as “The different peaks at BiVO4 electrodes”.
- “have” was revised as “has”.
- “most probably” was revised as “most probable”.
- “the kinetic limitation” was revised as “controlled by the kinetics”;
- “or possibly by-products of the oxidation adsorb on the electrode” was deleted.
- “was exhibited” was revised as “is shown”.
- “poor crystallization” was revised as “poor crystallinity”
Page 11:
- “highest electrochemical property were” was revised as “highest electrochemical signal was”.
- “3.4. Structure characterization of clavate BiVO4” was revised as “Structural characterization of clavate BiVO4”.
- “promote” was revised as “promotes”
- “depicting the uniform distributed of Bi, V and O elements, ” was revised as “depicts the uniform distribution of Bi, V and O elements”.
Page 12:
- “CV curves of on clavate BiVO4 electrode in 100 mM paracetamol with different scan rates of 1 to 480 mV s-1. Inset of (A) is a plot of the peak current against peak potential with different scan rates. Dependence of (B) log υ vs. log Ipa and (C) Ep vs. log υ at the clavate BiVO4 electrode. (D) DPV curves of clavate BiVO4 electrode with the concretion of paracetamol from 0 to 100 mM. Inset of (D) is the high concentration of paracetamol with 1-10 mM ranges.” was revised as “Fig. 5 (A) CV curves at on clavate BiVO4 electrode in 100 mM paracetamol with different scan rates from 1 to 480 mV s-1. Inset of (A) is a plot of the peak current against peak potential with different scan rates. Dependence of (B) log υ - log Ipa and (C) Ep - log υ at the clavate BiVO4 electrode. (D) DPV curves at clavate BiVO4 electrode with the concretion of paracetamol from 0 to 100 mM. Inset of (D) are DPV voltammograms corresponding to the high concentration of paracetamol with 1-10 mM ranges.”.
- “were recorded” was revised as “are shown”.
- “anodic” was revised as “anodic peak”.
Page 13:
- “diffusive mass transport” was revised as “diffusion-controlled mass transport”.
- “paracetamol detection were” was revised as “paracetamol determination are”.
- “compared with those of other morphologies BiVO4 electrodes” was revised as “compared with BiVO4 electrodes of other morphologies”.
- “affect the current response” was revised as “affect the DPV current response”.
Page 14:
- “A pill” was revised as “one tablet”.
- “Compare with” was revised as “Compared with”.
- “were exhibited” was revised as “are given”.
- “pharmaceutical analysis” was revised as “pharmaceutical tablets”.
Page 15: “on paracetamol detection” was revised as “for paracetamol detection”.

Reviewer 2 Report
Nanomaterials-1634265-Peer Review-r 1
Morphology Effect of Bismuth Vanadate on Electrochemical Sensing for the Detection of Paracetamol
The manuscript describes an electrochemical method to detect paracetamol. Various morphologies of BiVO4 were created using different additives and controlling pH. The clavate-shape BiVO4 demonstrated the greatest response in CV. The approach to verify the morphology effect of BiVO4 on its electrochemical response is original and interesting. The material characterizations and analyses were well performed and explained. The conclusion is aligned with the results and discussion of manuscript.
However, there are several issues that should be addressed in order for the manuscript to be published in Nanometerials Journal.
- Introduction should be re-written. The first paragraph describes the problems of paracetamol. But the last sentence doesn’t align with the rest of the paragraph. Likewise, the 3rd sentence in the 2nd paragraph is off base. The 2nd paragraph explains the rationale for using transition metals for paracetamol detection. Then, all of sudden, the 3rd sentence talks about catalysts. Please note that transition metals may be used as electrochemical catalysts for various applications, but they are not “catalysts”.
- The experimental section should explain how the BiVO4 powders were deposited and stabilized on an electrode. Also, what type of materials were used for the electrodes (working, reference, and counter)?
- An equivalent circuit diagram for CV should be presented.
- For the materials, the manufacturers and their locations should be presented.
- In the first sentence of the 2nd paragraph on page 10, (Fig. 3E) is in an incorrect place because it is not related to the content of sentence.
- DPV was used first time on page 9. The full name should be presented.
- Figure 6B is neither cited nor explained.
- There are numerous grammatical errors and typos including the subject-verb disagreement, misuse of singular and plural forms, misuse of definite and indefinite articles, and awkard sentences. These flaws diminish the quality of manuscript, and they must be corrected to be accepted for publication.
Author Response
Thanks very much for giving us the opportunity to revise the article. We have made changes to according to the reviewers' comments, and all the changes are highlight in manuscript.
Question 1: Introduction should be re-written. The first paragraph describes the problems of paracetamol. But the last sentence doesn’t align with the rest of the paragraph. Likewise, the 3rd sentence in the 2nd paragraph is off base. The 2nd paragraph explains the rationale for using transition metals for paracetamol detection. Then, all of sudden, the 3rd sentence talks about catalysts. Please note that transition metals may be used as electrochemical catalysts for various applications, but they are not “catalysts”.
Answer to Q1: Thanks for review’s kindly advise. The Introduction has been revised. And we change the word “catalyst” in manuscript.
Question 2: The experimental section should explain how the BiVO4 powders were deposited and stabilized on an electrode. Also, what type of materials were used for the electrodes (working, reference, and counter)?
Answer to Q2: Thanks for review’s suggestion. The preparation of BiVO4 electrode was supplied in manuscript.
“The BiVO4 electrodes with different morphologies were prepared by spin coating method. 5 mg BiVO4 powder was dispersed in 1 mL DI water by ultrasound for 10 min. Then the BiVO4 suspension was transferred to FTO substrates by spin coating. The as-prepared BiVO4/FTO electrodes were annealed in a muffle furnace with 200 oC for 2 hours for stabilization. Finally, the BiVO4/FTO electrodes with dif-ferent morphologies were obtained.”
The three-electrode system was also described in detail.
“The electrochemical experiment was performed using a conventional three-electrode system with the prepared BiVO4 as working electrode, a graphite rod as the counter electrode and a saturated Ag/AgCl electrode as the reference electrode.”
Question 3: An equivalent circuit diagram for CV should be presented.
Answer to Q3: Thanks for review’s suggestion. Cyclic voltammetry controls the electrode potential to scan repeatedly at different rates, and records the current potential curve. Generally, CV plot does not have the equivalent circuit diagram. The equivalent circuit diagram generally appears in AC impedance analysis for the purpose of research electrochemical interface behavior at the interface between electrode and solution. In AC impedance analysis, the double-layer capacitor, Helmholtz capacitance (CH), diffusion (Cdiff) capacitance, adsorption capacitance (Cads), resistance at double layer interface region (Rint), charge transfer resistor (Rct), adsorption resistor (Rads) and bulk solution impedance (Rbulk) and capacitance (Cbulk) should be taken into consideration in a real electrochemical double layer and its electrode/electrolyte interface model. The most common equivalent circuit diagrams are shown in the attachment.
Question 4: For the materials, the manufacturers and their locations should be presented.
Answer to Q4: Thanks for review’s suggestion. The manufacturers and their locations of reagent were supplied in 2.2 Chemicals and reagents.
“Bi(NO3)3.5H2O (99.0%,), NH4VO3 (99.9%), KCl (99.8%), NaH2PO4 (99.0%), Na2HPO4 (99.0%) and Na3PO4 (96%) were obtained from Aladdin reagent (Shanghai) Co., Ltd, China. HCl, Na2CO3 (99.5%) and NaOH (97%) were purchased from Macklin (Shanghai) Biochemical Technology Co., Ltd, China. The standard drug of paracetamol tablets (over the counter, OTC) was obtained from Taiji Pharmaceutical Industrial Co., Ltd, China.”
Question 5: In the first sentence of the 2nd paragraph on page 10, (Fig. 3E) is in an incorrect place because it is not related to the content of sentence.
Answer to Q5: Thanks for your kindly advise. Fig. 3E shows the CV curves of particle BiVO4 in 0.01 M PBS with and without 100 mM paracetamol. The particle BiVO4 is taken as s a reference sample, which can help us to understand the electrochemical performance of other morphologies.
We also changed the related content as “In order to research the influence of morphology on the electrochemical performance, the compared CV results of BiVO4 with clavate, fusiform, flowered, bulky, as well as the reference morphology of particle shape are show in Fig. 3A to 3E.” in manuscript.
Question 6: DPV was used first time on page 9. The full name should be presented.
Answer to Q6: The full name of DPV, differential pulse voltammetry, was presented on page 9.
Question 7: Figure 6B is neither cited nor explained.
Answer to Q7: Thanks very much for your advice. The Fig. 6B shows the mechanism of the electro-chemical oxidation and reduction of paracetamol. The related explanation has been added as “The mechanism of the electro-chemical oxidation and reduction of paracetamol is shown in Fig. 6B. The applied voltage promotes the separation of electrons and holes in BiVO4. Acetaminophen was rapidly oxidized on the BiVO4 electrode surface to form oxidation products, such as N-acetyll-p-quinoneimine.”
Question8: There are numerous grammatical errors and typos including the subject-verb disagreement, misuse of singular and plural forms, misuse of definite and indefinite articles, and awkard sentences. These flaws diminish the quality of manuscript, and they must be corrected to be accepted for publication.
Answer to Q8: Thanks very much for your suggestions. We have carefully revised the manuscript and all the changes are highlighted.

Round 2
Reviewer 2 Report
Nanomaterials-1634265-Peer Review-r 2
Morphology Effect of Bismuth Vanadate on Electrochemical Sensing for the Detection of Paracetamol
The revised manuscript improved significantly from the previous version. There are still a few minor errors that to be addressed in order to be accepted for publication in Nanomaterials.
Page 3:
- Semiconductor has been taken as an effective photocatalytic and .... sensor .... ---> Semiconductors have been taken as effective photocatalytic and .... sensors ...
- Therein transition metal oxide BiVO4, with an excellent .... ---> Therein a transition metal oxide such as BiVO4, with .....
- Medeiros et. al reported .... ---> Medeiros et al. reported ...
- .... electrocatalytic performance of the material. ---> .... electrocatalytic performance of metal oxides.
- ... the size and shape of the sample is ... to optimize its active area ... ---> ... the size and shape of them is ... to optimize their active areas ...
- ... efforts have ben made to engineer materials on ... ---> It is not clear whether the authors are saying any general “materials” or “metal oxides”. Please specify.
- ... on the nanoscale that has led to .... ---> ... on the nanoscale that have led to ...
- ... BiVO4 with larger surface ... ---> ... BiVO4 with a larger surface ...
- ... afford faster ... ---> ... afford a faster ...
Page 4:
- Scanning electronic microscopy ---> Scanning electron microscopy
- ... transmission electron microscope ... ---> ... transmission electron microscopy ...
- ... photoelectron spectrometer ... ---> ... photoelectron spectroscopy ...
- UV-vis ... ---> UV-Vis ...
- ... recorded at a ... ---> ... recorded with a ...
Page 5:
- Phosphate buffer saline ... ---> Phosphate buffered saline ...
- ... (Millipore) ... ---> ... (Millipore, Location) ...
- ... Na2CO3 were used as additive. ---> ... Na2CO3 were used as additives.
- ... ice ultra-pure ... ---> ... icy ultra-pure ...
Page 6:
- ... different morphologies BiVO4 ... ---> ... different morphologies of BiVO4 ...
- ... clavate morphology BiVO4 ... ---> ... clavate morphology of BiVO4
Page 7:
- Further change the pH of .... ---> A further change in the pH of ...
- ... lead to the dissolve or reshape ... ---> ... lead to dissolution or reshaping ...
Page 11:
- ... Kissinger et al ... ---> ... Kissinger et al. ...
Page 13:
- ... are show ... ---> ... are shown ...
Page 15:
- ... (Fig. 4A and B). Fig. 4D and 4E .... TEM image of ... shows ... ---> ... (Figs. 4A and B). Figs. 4D and 4E ... TEM images of ... show ...
Page 16:
- ... high index lanes ... ---> it is not clear what index lanes are. Shouldn’t it be index planes instead? Please check.
Page 17:
- Linear regression ... ---> A linear regression ...
- ... and Fe3+, et al. were added ... ---> it is not clear what it means by Fe3+, et al.
Page 18:
- Fig.6.B is presented, but never cited and explained in the body of text.
Page 19:
- ... high index lanes ... ---> Please see the comment above regarding it.
Author Response
Thanks very much for the meticulous revisions to the article. We have made changes to according to the reviewers' comments.
Question 1: ... high index lanes ... ---> it is not clear what index lanes are. Shouldn’t it be index planes instead? Please check. (Page 16)
Answer to Q1: Thanks for review’s suggestion. The high index lanes represent the higher the index plane in crystal. This is a relative concept. This concept is often used to indirectly reflect the interplanar spacing, and the interplanar spacing of the high-index plane is relatively large. In the book "Electrocatalysis" written by Sun Shigang, the crystal planes of face-centered cubic metals like platinum are classified, in which (111), (110), (100) at the vertices of the triangle are called basic crystals or low-index crystal planes. As long as one of (h, k, l) has an index greater than 1, it is called a high-index facet, which are distributed on the sides and interior of the triangle.
“The high index lanes refer to a facet with one of the indexes is greater than 1 in (h, k, l).”
Question 2: Fig.6.B is presented, but never cited and explained in the body of text (Page 18)
Answer to Q2: Thanks for review’s suggestion. We explained the mechanism of the electrochemical oxidation and reduction of paracetamol in manuscript.
“The mechanism of the electrochemical oxidation and reduction of paracetamol is shown in Fig. 6B. The applied voltage promotes the separation of electrons and holes in BiVO4. The oxidation peak of paracetamol could vary with the pH of the solution. Paracetamol converts to intermediate NAPQI easily at the pH of solution is 7.4. NAPQI can stably exist in the solution in a deprotonated form. The CV of paracetamol shows an oxidation peak and a relatively weak reduction peak. With the progress of the reaction, NAPQI gradually transforms into benzoquinone through other intermediates. The reduction peak of benzoquinone could be observed in CV.”
Question 3: ... high index lanes ... ---> Please see the comment above regarding it. (Page 19)
Answer to Q3: Thanks for review’s advice. The concept of high-index crystal planes has been explained in manuscript.
